# Latest Advances in Protein-Recovery Technologies from Agricultural Waste

**DOI:** 10.3390/foods10112748

**Published:** 2021-11-09

**Authors:** Farhana Iylia Fatinee Mohd Yusree, Angela Paul Peter, Mohd Zuhair Mohd Nor, Pau Loke Show, Mohd Noriznan Mokhtar

**Affiliations:** 1Department of Process and Food Engineering, Faculty of Engineering, Universiti Putra Malaysia, Serdang 43400, Malaysia; iyliafatinee23@gmail.com (F.I.F.M.Y.); noriznan@upm.edu.my (M.N.M.); 2Department of Chemical and Environmental Engineering, Faculty of Science and Engineering, University of Nottingham Malaysia, Semenyih 43400, Malaysia; angelapaulpeter@gmail.com; 3Laboratory of Halal Science Research, Halal Products Research Institute, Universiti Putra Malaysia, Putra Infoport, Serdang 43400, Malaysia

**Keywords:** agricultural waste, protein, amylase, liquid biphasic system, separation

## Abstract

In recent years, downstream bioprocessing industries are venturing into less tedious, simple, and high-efficiency separation by implementing advanced purification and extraction methods. This review discusses the separation of proteins, with the main focus on amylase as an enzyme from agricultural waste using conventional and advanced techniques of extraction and purification via a liquid biphasic system (LBS). In comparison to other methods, such as membrane extraction, precipitation, ultrasonication, and chromatography, the LBS stands out as an efficient, cost-effective, and adaptable developing method for protein recovery. The two-phase separation method can be water-soluble polymers, or polymer and salt, or alcohol and salt, which is a simpler and lower-cost method that can be used at a larger purification scale. The comparison of different approaches in LBS for amylase purification from agricultural waste is also included. Current technology has evolved from a simple LBS into microwave-assisted LBS, liquid biphasic flotation (LBF), thermoseparation (TMP), three-phase partitioning (TPP), ultrasound-assisted LBS, and electrically assisted LBS. pH, time, temperature, and concentration are some of the significant research parameters considered in the review of advanced techniques.

## 1. Introduction

Agricultural waste is residual material generated from various agricultural activities, such as coffee pulp from the coffee industry, husks from the cereal industry, and peels from the starch-based industry [1]. Every year, a large amount of these residues will be produced from agricultural activities, which will eventually deteriorate and have a detrimental effect on agricultural resources, human and animal health, as well as the environment if no appropriate action is taken to manage the waste [2]. The current intensive agricultural activities are predicted to generate a large volume of residues, with an annual production of 998 million tons of agricultural trash [3].

Nowadays, agricultural waste has become an indisputable requirement of rural ecological civilization construction to mitigate the negative impacts that threaten long-term development and human health. Agricultural waste recycling and usage are important in boosting modernized agriculture growth [4]. One of these efforts involves recovering proteins bound in the waste. These proteins have a paramount value in the food, chemical, and pharmaceutical industries. Research on the recycling and utilization of agricultural industry waste has received much attention in the recovery of proteins and their modification during processing [5,6].

Protein is the link of amino acids bonded by peptide bonds forming the primary, secondary, tertiary, and quaternary structure of protein [7]. Regardless of the superiority of the protein quantity in animals compared to plants, plant-derived protein has become one of the alternative options in overcoming the increasing environmental issue caused by animal-protein production. Plant-derived protein is now receiving attention for its functionality as a sustainable protein source [8]. This explains the amplitude of research on plant-derived protein, specifically the recovery of enzyme protein from agricultural waste [9,10,11,12].

Amylase is a type of protein with activation sites that can be derived from plants. Amylase is recognized as one of the main industrial enzymes, accounting for over 30% of the global enzyme industry and is mostly used in the food, fermentation, and pharmaceutical industries. [7,8]. Previously, the large-scale production of this enzyme was limited to only certain strains of bacteria and fungi, making them the only potential resources to meet the huge demand of the industries [9]. Nevertheless, the extraction of amylase from the plant source is gaining more attention from the current biotechnology industry as it is lower in cost and toxicity [10]. This opens more avenues for the use of agricultural waste in amylase production.

Amylase purification can be completed either by various conventional or advanced methods available in today’s technology [13,14,15,16,17,18,19]. Some of the prominent conventional methods for the extraction and purification of enzymes are filtration, membrane extraction, precipitation, liquid–liquid extraction, ultrasonication, and chromatography, which are described in this review article, as well as advanced methods, including the liquid biphasic system (LBS), liquid biphasic flotation (LBF), thermoseparation (TMP), and three-phase partitioning (TPP). An efficient downstream processing method is needed to complete a large-scale purification of enzymes and protein to preserve their biological activity [20,21].

The idea of LBS technology as an analytical separation technique was established in 1960 with the concept of mixing two different polymers, which resulted in an aqueous medium with two separate phases [22]. Generally, the concept of this technique involves the acclimatization of the biomolecules of interest, either to the top or bottom phase, once a physicochemical interaction is formed by the phase-forming components [23]. Multistage processes, longer operation periods, complicated routes, extra cost, and large energy inputs in the recovery and extraction processes have been shown to be solved using an LBS. Further development of this conventional polymer-based LBS has allowed the emergence of advanced technologies integrated with LBS. This integrated LBS technique is designed to specifically fit the need for extracting various biomolecules. Conventional LBS issues involving high electrolyte concentration biomolecules, costly phase-forming components, and a highly viscous system can be overcome by these advanced technologies assisted by LBS [23].

The implementation of conventional and advanced methods is reviewed in this article. The highlighted techniques are based on the LBS for agricultural waste. Furthermore, the differences between the conventional LBS and the advanced liquid biphasic electrically assisted system are also explained.

## 2. Extraction of Amylase from Agricultural Waste

### 2.1. Characteristics of Agricultural Waste and Protein

Agricultural and industrial waste can be separated into two categories: agricultural and industrial residues. Agricultural residues include stems, stalks, leaves, seed pods and husks, seeds, roots, bagasse, and molasses, as well as field and processing residues. Meanwhile, industrial residues are mostly from the food-processing industries, such as potato peel, orange peel, soybean cake, cassava peel, and other organic residues.

Industrial residues are expected to multiply in tandem with the increasing population and demand in food supply [1]. This corresponds to the development of the high-input agricultural trend, which will improve the overall residue production, including agricultural waste, by 1.3 Pg dry matter per year [13]. However, these protein-rich residues have started to gain interest for their economically attractive value and capability to be recovered. The residues are now mostly used for the extraction and utilization of usable protein and applied in foods and supplements [14]. This agricultural waste should be significantly regarded as a potential resource to cope with the modern food-technology process and in line with a complete life cycle analysis system [2].

Protein is naturally synthesized in plants and animals; generally, protein is abundant in animals compared to plants [8]. Hicks and Verbeek (2016) stated that the growing worldwide demand for animal-based products necessitates a significant rise in plants and other feed resources, resulting in a much higher amount of protein-rich materials being generated as waste than the protein supplied for consumption. The major facet of this occurrence is to convert these agricultural wastes into usable protein [14]. The discovery of usable protein from these wastes will be feasible along with the technology available for recovering nutrient-rich protein. Membrane separation, adsorption, microbe-assisted protein recovery, and other conventional extraction methods have been presented as potential strategies for protein recovery from waste [15,16,17]. The recovery of enzyme protein is one of the concerted efforts for converting these wastes into usable protein in the industry [9,18,19].

Amylase is recognized as a crucial industrial enzyme protein, comprising approximately 30% of the world enzyme market [20,21]. It is eminent for the food, fermentation, and pharmaceutical industries. Amylase can be found in animals, bacteria, and plant cells. Despite various sources of amylase, only fungi and bacterial amylase dominate the industrial sector. Previously, large-scale production was limited to only certain strains of bacteria and fungi (extracellular protein), making them the only resources susceptible to meet the huge demand of the industries [22]. However, the discovery of biotechnologies has found that plants (intracellular protein) can suffice as a rich source of plant-derived enzymes for biotechnological and industrial purposes at lower cost and toxicity [9].

#### 2.1.1. Presence of Enzymes in the Agricultural Waste Stream

Enzymes are proteins that behave as biological catalysts in a series of biological reactions. They increase the pace of reaction by lowering the activation energy, which helps to reduce the cost of manufacturing in terms of resources required. Enzymes have been widely used for ages to produce food such as yogurt, wine, and cheese. However, conventional methods of industrial enzyme production overlooked the production cost due to the fermentation media and complex processes [23]. Hence, to overcome the drawbacks of conventional methods and reduce the cost of production whilst fulfilling the industrial demand, various agricultural wastes have been assessed for the extraction and purification of enzymes [23]. Therefore, the conversion of renewable resources due to the growth of the agro-industrial sector has attracted the interest of researchers for decades, witnessing the increase of studies on various techniques for enzyme production from agricultural waste [23].

Enzyme extraction from agricultural waste has long been discovered due to its potential industrial applications. The ability to reduce production costs and improve enzyme performance for industrial purposes is greatly enhanced by the valorization of agricultural waste. Actinobacterial enzymes produced economically from agricultural waste as an alternative in utilizing the biomass generated as waste have been studied, where amylase, cellulase, tannase, xylanase, protease, and laccase are among the enzymes produced from the biomass generated [24].

Bromelain or plant protease is reportedly present in pineapple peel, core, crown, and leaves [25]. The highest proteolytic activity is discovered from the extract of pineapple crown. Bromelain possesses broad purposes in industrial applications, such as tenderization, foods, detergents, and the textile industry. However, bromelain extraction becomes an issue because the growth of pineapple crops is mostly designated for food production. Thus, the variability of agricultural waste from pineapples (crown, peel, stem, and core) and the capability for bromelain enzyme extraction will cater to the demand of this plant protease for industrial use [25].

#### 2.1.2. Classification of Amylase

Amylase can be found in the plant, microbial, and animal kingdoms. Amylases are enzymes that break down starch by catalyzing the hydrolysis of α-1-4-glycosidic linkages in alpha polysaccharides. For millennia, plant-derived amylase has been widely used in the brewing industry, whilst fungiform amylases are commonly utilized in the production of oriental delicacies [26].

Amylases can be divided based on branches. The first type of amylases consists of hydrolases, endoamylases, and exoamylases, and the second type of amylases comprises alpha-amylase, beta-amylase, and gamma-amylase, as presented in Table 1 [27]. Alpha-amylase (α-1,4-glucan-glucanohydrolase, EC 3.2.1.1) is classified as an extracellular enzyme for degrading α-1,4-glycosidic linkage of starch into oligosaccharides or saccharides [28].

Beta-amylase (1,4-D-glucan maltohydrolase; glycogenase; saccharogen-amylase, EC 3.2.1.2) catalyzes the hydrolysis of the second α-1,4-glycosidic linkage by cleaving the linkage from the non-reducing end. During the ripening of fruits, beta-amylase converts starch to maltose, which gives matured fruits their sweet flavor.

Gamma-amylase (EC 3.2.1.3) has the alternative names of glucan-1,4-α-glucosidase, amyloglucosidase, exo-1,4-α-glucosidase, glucoamylase, lysosomal α-glucosidase, and 1,4-α-D-glucan glucohydrolase. This enzyme breaks the α-1,6-glycosidic linkage and the last α-1,4-glycosidic linkage at the non-reducing end of amylopectin and amylose.

#### 2.1.3. Presence of Amylase in Agricultural Waste Stream

Amylases have tremendous applications in the industrial sector. Hence, to meet the demand for amylases, various technologies and agricultural wastes have been explored to achieve a high yield of amylase through simpler steps and low-cost processes.

Sagu et al. (2016) [29] carried out the purification of β-amylase from the stem of *Cadaba farinosa*, a multipurpose shrub commonly found in West Africa, using TPP. This plant is well-known for its pharmacological properties, where the young leaves, flower buds, and roots are used as sources of drugs [26]. It was reported that this novel experimental design for the extraction of β-amylase from the stem of *C. farinosa* yielded a purification factor of 11.18 with an enzyme activity recovery of 51.1% [30]. Hence, the potential of *C. farinosa* waste for the extraction of amylase is verified.

Amid et al. (2014) evaluated the extraction of amylase from dragon fruit peel [31]. Dragon fruit is recognized as one of the world’s leading commercial tropical fruits. It carries around 33% of the peel weight of the overall fruit weight, which is typically removed during processing, especially in the beverage processing industry [32]. The isolated amylase demonstrated high stability at low pH, thermostability, and surfactant agent stability. Furthermore, the amylase extracted from dragon fruit peel was found to be a viable low-cost enzyme for use in industrial and biotechnological applications [31].

In research carried out by Lawal et al. (2014) [33], a soil sample from the cassava peel dumpsite was taken to obtain the pure culture of *Aspergillus niger.* The best strain of *A. niger* was collected to perform enzymatic extraction and purification. This experiment concluded that cassava peel is an adequate substrate for amylase production. It is also reported that an innovative enzyme technology to utilize the abundance of cassava peel is readily developed by the Federal Institute of Industrial Research, Oshodi, Nigeria [23]. Hence, the extraction of this enzyme from cassava peel waste has further proven the presence of the enzyme in agricultural waste.

The utilization of several types of agricultural wastes as substrates for amylase production was investigated. Soybean meal, wheat bran, maize protein, hazelnut cake, and whey were used as natural substrates to promote α-amylase synthesis in *B. amyloliquefaciens*. From the study, the medium cultivated with soybean meal produced the most amylase. The evidence from this experiment highlights the importance of amylase in the industrial process, as well as its ability to utilize agricultural waste [34].

#### 2.1.4. Application of Amylase

Amylase is mostly used in industrial applications for the brewing and baking industry. In alcohol brewing, starch-based plants (e.g., cassava and tapioca starch) are converted to sugar by enzymatic fermentation. Bread is allowed to impart and rise when starch is converted to simple sugar. This enzyme is distributed and commonly used in numerous biotechnological and industrial applications [7].

One of the applications of amylase is in the industrial production of glucose and fructose from starch. In this process, starch is mixed with water and heated to dissolve starch granules. Amylose and amylopectin are broken in water in the traditional technique of sweetener processing. Next, the mixture is treated with acid until the desired degree of hydrolysis is achieved. This heated slurry is passed to an enzymatic reactor for further enzymatic treatment. After, the mixture is further reacted with a saccharifying enzyme to break down dextrin. This saccharification process uses glucoamylase to hydrolyze the α-1,4-glycosidic bonds from the non-reducing end of the chain. Nowadays, acid hydrolysis is replaced with enzymatic hydrolysis, mainly α-amylase, due to the drawback of using acid during the operation.

The industrial production of cyclodextrin also uses amylase enzymes. Cyclodextrins are used in medicines; agricultural chemicals; and cosmetic, pharmaceutical, and food applications. First, starch hydrolysate containing cyclodextrin is prepared by adding CGTase with a gelatinized starch solution that is heated and prepared by liquefying alpha-amylase beforehand. The mixture is filtered, and β-cyclodextrin from the hydrolysate solution further reacts with the saccharifying enzyme. This saccharified slurry is purified to obtain cyclodextrin from the sugar solution [19].

### 2.2. Conventional Methods in Extraction and Purification of Protein from Agricultural Waste

The huge amount of agricultural waste generated per year eventually leads to pollution and economic loss. However, these waste streams are rich in bioactive compounds, including proteins that have the potential to be recovered and re-utilized as functional foods, medications, cosmetics, and biopackaging [28]. Hence, much research has been completed on the recovery and utilization of these agricultural wastes [28,35,36,37,38].

#### 2.2.1. Membrane Extraction

Membrane extraction is a bioseparation method that uses a membrane, which is an interphase that is usually heterogeneous. The membrane acts as an obstacle to the flow of molecular and ionic groups present in the liquid or vapor phase. Membranes have traditionally been used for size-based separation with high throughput but low-resolution requirements. This method includes microfiltration, ultrafiltration, and nanofiltration techniques for protein concentration and buffer interchange [17]. Separation using membrane extraction has been proposed for enhancing processes due to the absence of chemical usage, reduced energy consumption, and simpler scale-up of operations [39].

Membrane technology has been demonstrated to be excellent in recovering proteins from waste sources [15,40]. The ultrafiltration method was applied to extract valuable protein from poultry-handling industry waste [15]. The result obtained from this experiment demonstrated the highly relevant and ideal application of ultrafiltration to extract protein from the poultry waste stream.

Amylase separation via microfiltration was reported by Rodrigues et al. (2017) [41]. Crude enzyme extract was subjected to a microfiltration process to separate enzymes of different molecular weights from other low-molecular-weight compounds of the previous fermentation process. It was determined that the enzymatic activity of the retentate increased by 38% [41]. This is due to the removal of lower-molecular-weight compounds from the retentate, which could affect amylase activity [41]. The efficiency of a two-stage membrane technique for purifying protease from a pineapple waste mixture was evaluated [42]. Two-stage ultrafiltration, enzymatic pretreatment, and diafiltration were included in this system. During membrane filtration, the enzymatic pretreatment improved flux performance. The excellent membrane selectivity of the diafiltration procedure helped this system to achieve a 4.4-fold increase in enzyme purity [42].

Protease purification was conducted via a one-step ultrafiltration process of feather meal [43]. Keratinolytic proteases were produced by *Bacillus* sp. P45 with chicken feather meal as the substrate. The optimized ultrafiltration process recorded an enzyme recovery of 87.8% with a 4.1-fold purity increment. The limitation of this membrane-based extraction is flooding and loading limits while passing through the membrane, emulsification on the membrane, and large solvent inventory and high investment in the machinery [44,45]. On the other hand, the benefit of membrane separation techniques is energy efficiency, where most of the membrane-based extraction techniques require low energy consumption and possess small dimensions that can reduce space consumption [45].

#### 2.2.2. Precipitation

One of the most frequent methods used for separating enzymes and proteins is via precipitation. The idea behind this procedure is to introduce salts, polar and non-polar solvents, and organic polymers into cell extracts by changing the temperature or pH [25]. The concept of precipitation is the phenomenon whereby a substance is dissolved in a solution, which will then emerge, and the emerging precipitate will be separated from the solution [46]. For example, for the precipitation of protein by acid, the changes in the pH of the medium affect protein structure. When acid is added, the hydration sphere surrounding the protein is disrupted, leading to precipitation [47].

Haslaniza et al. (2014) applied the precipitation method to optimize the production of protein hydrolysate from cockle meat wash water [48]. This experiment demonstrated precipitation as a reliable technique in recovering protein from waste as the objective of this experiment was achieved. Furthermore, the optimized parameter was successfully identified [48].

A study reported the separation of amylase from agricultural waste by precipitation. Ammonium sulfate precipitation was used to separate amylase from cultivated dhal industrial waste with 40% activity [49]. The same technique was also used to purify amylase from cultivated tapioca liquid waste, which managed to obtain the enzyme-specific activity of 37.56 ± 0.38 U/mg [50].

The advantages of precipitation include the possibility to be applied in a continuous process due to its simplicity, scalability, proven usage in bioprocessing, as well as good biocompatibility and storage stability [46]. Meanwhile, the drawbacks of precipitation are the possible introduction of additional impurities, and quality control is needed because the yield of separation may be significantly influenced by pollutant ions, such as divalent ions [46].

#### 2.2.3. Ultrasonication

Ultrasound-assisted extraction involves several mechanisms, including fragmentation, erosion, capillarity, texturization, and sonoporation [51]. This technique can be performed by bath or probe modes. A comparison between ultrasound-assisted extraction and conventional solvent extraction was made to purify carotenoids from pomegranate peel [52]. They concluded that ultrasound-assisted extraction exhibited superior attributes in terms of a green process, low energy consumption, and safer processing with a higher yield of carotenoids.

Ultrasonication is regarded as one of the green technologies for the extraction of plant-based proteins [53]. Ultrasound-assisted extraction was conducted on wampee seed to extract protein, which is regarded as an advantageous nutraceutical and food ingredient. The finding from this experiment proved that ultrasonication is feasible in recovering protein from wampee seed [54].

Jain and Anal (2016) carried out ultrasound-assisted extraction of functional protein hydrolysates from the membrane of chicken eggshells [40]. According to this study, ultrasonic treatment is an effective method for separating protein hydrolysates.

Another study on ultrasonication was performed for the extraction of bioactive compounds from amaranth [55]. Amaranth is an underutilized plant with numerous precious bioactive compounds, such as polyphenols, betaxanthins, and betacyanins, that can inhibit fatal diseases [56]. Based on the study performed by Ahmed et al. (2020), ultrasound-assisted extraction enhanced the extraction of bioactive compounds and the antioxidant activities of amaranth [55]. This method also has the potential to serve as an alternative to the conventional extraction technique [55].

The application of ultrasound-assisted extraction is reported to be more efficient and offers a shorter extraction time with lower solvent utilization; furthermore, there is a huge potential for the automation of this extraction process [55,57]. Meanwhile, the drawback of ultrasound-assisted extraction is the presence of oxidation that can cause negative impacts on bioactive compounds, the decrease in nutritional content, and the occurrence of cell rupture on the extracted bioactive compounds [55].

#### 2.2.4. Chromatography

Chromatography can be classified into four separation techniques: ionic exchange, surface adsorption, partition, and size exclusion [58]. Theoretically, the separation of molecules occurs between the movement of the mobile phase and the stationary phase.

Melnichuk et al. (2020) reported the valorization of two types of agricultural waste (i.e., soybean and wheat) by solid-state fermentation [19]. Size-exclusion chromatography was used to purify the enzyme protein, α-amylase, for the fermented samples. The recovery and purification factors of 83% and 6 were achieved, respectively.

Other research was conducted using chromatography for the purification of lithium [59]. Lithium is an alkali metal that is highly reactive, contains highly soluble salts, and possesses a low ionic charge [60]. Chromatography recovered lithium at a higher efficiency than extraction using the conventional liquid–liquid extraction (LLE) method [59].

Chromatography solves the drawbacks of LLE, such as the high consumption of solvent and complex extraction stages, by achieving higher purification of lithium [59]. Nevertheless, research on extraction using chromatography is still in premature development and operated on a small laboratory scale [59].

#### 2.2.5. Liquid–Liquid Extraction

When two immiscible or partially soluble liquid phases are stacked together, LLE is used to separate components from one phase to the next. Some of the most common varieties of this bio-separation technology principles include an aqueous two-phase micellar system, an aqueous two-phase polymer system, and a two-phase reverse micellar system [61,62]. When water-soluble polymers are mixed with other polymers, solvents, or inorganic salts at quantities above their critical concentrations, a two-phase system emerges. This principle is used to separate, concentrate, and fractionate biological solutes and particles, such as proteins, enzymes, and nucleic acids.

The LLE process is based on the movement of the targeted analyte from one phase to another when two immiscible liquids come into contact. This method is theorized to be dependent on preferential mass transfer and dissolution of target biomolecules in a complex aqueous matrix. The thermodynamic forces will drive the transfer of chemical species across the phase. The traditional method made use of a separatory funnel to quantitatively transfer the liquid out. Hence, relative density plays an important role, where the less-dense organic phase will reside in the top phase and the aqueous sample in the bottom phase [63].

Liquid–liquid extraction is utilized in the industry, particularly in the chemical and mining industries, as well as in the recovery of fermentation products at the end of the process [64]. Along with its simplicity, low cost, and ease of scaling up, LLE has been used to purify biomolecules on a large industrial scale for more than a decade [61]. The advantages of employing LLE include low viscosity, lower mechanical cost, and shorter time of phase separation. Most crucially for industrial applications, LLE has been certified by industry regulatory bodies [65].

By considering the benefits offered by this technique, researchers have explored its potential for improvements in terms of separation mechanism and setup. Hence, advanced separation techniques have been introduced by following the basic principles of LLE. These techniques are discussed in the next subtopic.

### 2.3. Advanced Liquid–Liquid Extraction Techniques in the Recovery and Purification of Proteins from Agricultural Waste

Several advanced techniques have been developed based on the separation principles of LLE. These techniques aim to overcome some drawbacks of the conventional LLE method by integrating several technologies to improve separation efficiency and process feasibility. These techniques have been reported to be successfully applied for the separation of proteins from agricultural waste. Advanced LLE techniques have received positive feedback for the application of LBS, LBF, TMP, and TPP on the extraction of proteins from agricultural waste [29,66,67,68].

#### 2.3.1. Liquid Biphasic System

An LBS is a popular approach for separating proteins, including enzymes and antibodies. Due to weak organic stability and the potential of protein denaturation in organic solvents during separation, the previously established conventional LLE employing an organic aqueous phase system is no longer a viable bioseparation approach [69]. Thus, complex fluids have been studied to solve the drawbacks of conventional LLE, resulting in studies on the aqueous two-phase system, which provides a biocompatible environment for the separation of bioactive compounds [61]. The LLE method employs a water–organic solvent two-phase system, whereas an LBS employs water-soluble polymers combined with either polymer or inorganic salts at a critical concentration, which explains the biocompatibility difference between LBS and LLE [61].

In general, the notion of LBS is generated by combining two incompatible polymers or combining a polymer solution with inorganic salts at a threshold concentration until two immiscible phases are developed, hence the name “LBS”. The polymer will dominate either side of the phases, or one side will be dominated by polymer and the other by salt, where the distribution of biomolecules in the phase is influenced by several factors, such as pH, polymer molecular weight, and salt type [11]. The targeted bioactive compound will be bound to one of the phases in this system. The partitioning of the molecules and the phase formed in an LBS are shown in Figure 1.

An LBS was applied to purify the enzyme protein, thermo-acidic amylase, from red pitaya peel, where an organic solvent and a thermoseparating polymer were used in the system [31]. The recovery and recycling of the components were observed at each successive step of the system. A satisfactory purification factor of 14.3 and a high yield of 96.6% with the recovery and recycling of copolymer at a rate above 97% were obtained. These results proved that the system is more economical compared to conventional LLE due to its recovery and recycling efficiency.

Another study on the application of LBS was conducted for the purification of α-amylase from the cultivation of *Bacillus subtilis* by the LBS [70]. A two-fold purification factor with over 90% amylase yield was achieved at the optimized conditions from the experimental model. This indicates the excellence of LBS in partitioning bioactive compounds.

An LBS was applied for the extraction of polyphenol oxidase and bromelain from pineapple [71]. Different partitioning of bromelain in the top phase and polyphenol oxidase in the bottom phase was accomplished using the LBS. At optimum conditions, bromelain was recovered at 228% yield with a 4.0-fold purification factor, whereas polyphenol oxidase was recovered at 90% yield with a 2.7-fold purification factor. This finding signifies the excellence of LBS to purify bioactive compounds from pineapple.

#### 2.3.2. Liquid Biphasic Flotation

Liquid biphasic flotation is the incorporation of conventional LBS and the principle of solvent sublation (SS) with the presence of bubbles [72]. The SS process is based on bubble-separation technology, where aqueous hydrophobic chemicals are adsorbed onto the bubble surface of the ascending gas stream bubble and transferred to the immiscible top phase. The incorporation of LBS and SS involves the liquid medium of the LBS phases to promote the mass transfer of biomolecules from the SS system, thus improving the efficiency of the phase formation of the immiscible liquid [67].

An LBF system was applied for the extraction of protein from expired dairy products [73]. In this study, the final protein recovery and the separation efficiency were 94.97% and 86.29%, respectively. The findings highlight a great potential of an LBF system in reusing the recycling phase component for the subsequent extraction process. Jiang et al. (2019) [74] used the LBF method to purify and characterize ovalbumin from salted egg white. The results in this study indicated that the purified ovalbumin was at a satisfactory state with no substantial differences in terms of the protein structure between the LBF method and the conventional method [74].

A study on the isolation and fortification of antioxidant peptides from whey protein isolate hydrolysate was carried out using both LBS and LBF systems [74]. It was concluded that both systems allowed the purification of peptides in a simple, fast, and inexpensive manner. Meanwhile, LBF provides better selectivity, scale-up, process integration, continuous operation, and high throughput in mixture separation compared to a standard LBS.

The separation efficiency of 82.67% and yield of 80.67% were achieved during the integrated fermentation and recovery of lipase from *Burkholderia cepacia* via the LBF system [75]. The fermentation of lipase from *B. cepacia* was integrated with the extraction of lipase from the fermentation broth via the LBF system, followed by separation and purification by a similar system. The application of LBF on the upstream fermentation process allowed bacteria to grow faster and produce higher yields compared to conventional fermentation, followed by the subsequent downstream separation by LBF, producing a lipase yield of 80.67%. The researchers claimed that employing the LBF system for fermentation and combining upstream and downstream processing in a single system has proven to accelerate product formation, boost product output, and facilitate downstream processing.

#### 2.3.3. Thermoseparation

Thermoseparation is basically an LBS with the involvement of a temperature-stimulated phase separation, known as cloud point extraction (CPE). CPE is linked to the solubility of thermoseparating polymers, which will decrease when the temperature of the system of the aqueous solution increases. Frequently used thermoseparating polymers include the diblock and triblock copolymers of hydrophilic ethylene oxide (EO) and hydrophobic propylene oxide (PO), which is also called EOPO. This advanced temperature-induced LBS emerged due to the limitation of conventional LBS, which encountered difficulties in recycling the phase-forming components [76].

A thermoseparating, aqueous, two-phase system was used to separate alkaline proteases from fish viscera using different parameters [77]. The integration of the bottom salt-rich phase and the bottom EOPO-rich phase with a ratio of 0.5:1.5 (*w*/*w*) during the recycling stage resulted in the best recovery and purity. When the EOPO polymer was utilized, the separation system yielded a total recovery of 91.62%. The study indicated that using a TMP LBS would allow for a quick and cost-effective process, as well as low energy consumption, environmental friendliness, and ease of operation upscaling.

Show et al. (2012) [78] reported the purification of lipase from *B. cepacia* by the thermoseparating LBS method. The use of EOPO allows effective recycling of the two phases formed in this system, where the top phase is the harvested products that are depleted by EOPO, and the bottom phase is the concentrated EOPO and cells with the potential of being reused in fermentation. *B. cepacia* lipase was purified in a single step with a 99% yield at optimized conditions. This study manifested that extractive fermentation via LBS possesses the potential to refine lipase production and recovery using thermoseparating polymers.

#### 2.3.4. Three-Phase Partitioning

The concept of a TPP system is established from the combination of inorganic salt with a crude extract containing protein and tert-butanol. The presence of tert-butanol and inorganic salt will aid the recovery of protein from the mixture solution. The hydrophobic part of protein will be bound with tert-butanol, reducing the density of protein molecules. Hence, these molecules will move to the surface of the denser mixture [79,80,81]. This technique is widely used to purify various enzymes and proteins with high purity and recovery. As shown in Figure 2, aqueous ammonium sulfate and crude extract were added with tert-butanol and left to stand for an hour before organic phases were formed (i.e., a protein or enzyme precipitate phase and an aqueous phase), producing three layers of phases.

Gagaoua and Hafid (2016) [80] studied the purification of different types of proteases in the TPP system. They concluded that the behavior of the phases formed is dependent on the molecular weight and the isoelectric point. Despite the availability of various separation and purification techniques, TPP is a better alternative extraction method that requires fewer steps and simpler procedures.

The behavior of proteases from papaya peel extracted by the TPP method was studied by Chaiwut et al. (2010) [10]. The optimized conditions for the extraction of proteases achieved 89.4% recovery and a 10.1-fold purify increment. This proves the efficiency of the TPP system for the purification of proteases from papaya peels.

#### 2.3.5. Integration of LBS with Other Technologies

The integration of LBS with other extraction technologies to enhance the separation process has been receiving great interest among researchers nowadays. For example, the integration of LBS with microwave, ultrasound, and electrical systems has been reported to improve separation efficiency. Microwave-assisted extraction is a method that evenly heats the solvent trapped inside sample pores to ease energy penetration into the pores, as the solvent can absorb the microwaves. The microwave energy will result in ionic conduction and dipole rotation of each molecule, eventually heating the solution [82]. The integration of an LBS with microwave heating will intensify the effect of microwave extraction and a two-phase extraction system in a one-step procedure, resulting in a rapid increment of temperature, enhanced diffusion of the solvent into the matrix, and assisted dissolution of components into the solvent [82,83,84]. For instance, microwave-assisted extraction was completed to extract rice bran protein. In comparison to the conventional method of extraction (alkaline extraction), the protein yield of microwave-assisted extraction was higher than the amount of protein extracted by the alkaline extraction [85]. This illustrated the potential of microwave-assisted extraction integrated with LBS to be reliable and effective in extracting protein.

Ultrasonication is one of the mechanical approaches for cell disruption. The principle of ultrasonication-assisted LBS is by installing ultrasound waves into the LBS system. The ultrasonic waves passing through the liquid provide micros-movement, which enhances the cell disruption. Ultrasonic waves have a powerful and high impact on solid surfaces, breaking cell membranes and allowing biomolecules to be released. They also have the potential to induce the penetration of solvent into cellular materials, which will enhance the mass transfer between the solvent and cellular materials [86]. This treatment is beneficial in terms of energy savings, greener processing, reduced operating cost, and is suitable for large-scale processing and effective cell disruption for the extraction of biomolecules. In comparison with conventional extraction methods, ultrasound-assisted extraction is safe and effective, based on prior investigations [87]. The incorporation of ultrasound with the LBS will increase the extraction yield, where the approach becomes more efficient and economical due to the capability to allow the integration of extraction and separation of bioactive constituents in a one-step process [88]. An experiment was completed to extract protein from Bombay locusts by implementing ultrasound-assisted extraction. The protein yield from the ultrasound-assisted extraction was significantly higher compared to the typical conventional extraction [89]. This foresees the prospective of LBS integrated with ultrasound-assisted extraction for the recovery of protein as appealing and worthy of further research.

An LBS integrated with an electrical system has been one of the current studies in line with the development of a green and effective bio-separation technology in the downstream bioprocessing industry [90]. Electrically assisted LBS is an extraction technique involving mild cell disintegration using electrical treatment. The pulsed electric field (PEF) is an electrical treatment that rearranges the membrane, where the cell membrane is charged with short electrical pulses until the membrane is sufficiently disrupted, resulting in pore development. The combination of this electrical treatment with an LBS will benefit the efficiency of the extraction [91]. The setup of this technique involves the dipping of two graphite electrodes into the LBS or LBF system (i.e., an anode and a cathode), where the anode detects the current flow while the cathode controls the voltage applied to the system, as shown in Figure 3a,b. Previously, an experiment for the recovery of protein from *Chlorella vulgaris* by using electrically assisted extraction was completed [92]. The satisfactory amount of protein recovered during this research anticipates the potential of LBS integrated with an electrical system to serve the recovery of protein.

Despite the potential for better separation efficiency offered by these advanced techniques, the studies on protein recovery are still insufficient. This statement explains the lack of explanation on advanced LBS methods (microwave-assisted LBS, ultrasound-assisted LBS, and electrically assisted LBS) for protein recovery in this article. Further studies are required to determine the effect of these techniques, particularly on the protein structure. It is expected that at a higher level of microwave energy, ultrasound intensity, and electrical pulse, protein denaturation might occur. The increasing flow of microwave energy through the system will affect the 3D protein structure due to the breakdown of the hydrogen bonds and sulfur bridges [93]. Similarly, the unfolding of proteins might occur due to the high intensity of ultrasound that leads to a large increase in local temperature and pressure [94]. Meanwhile, Jiang et al. [95] reported the structural transitions of the protein conformation induced by the increasing strength of the electric field. Hence, extensive optimization works are an interesting venue in which to venture further to find a balance between a good separation efficiency with minimal impact on the protein structure for these integrated LBS techniques to be fully applied for protein separation.

### 2.4. Comparison of Conventional and Advanced LLE Techniques

Despite the simplicity of the operation and apparatus setup of conventional LLE methods, the drawbacks of conventional methods are significant, making the recovery and extraction of targeted samples less favorable with the emergence of the latest and advanced LLE methods [83,90,91]. The comparison of conventional and advanced LLE methods is presented in Table 2.

First, in terms of environmental impact, conventional LLE requires the use of large volumes of highly pure solvents, where the reagents employed are no longer usable or recyclable and must be dumped into the environment, resulting in negative environmental consequences. This drawback is overcome by advanced LLE methods, where the phase-forming components are non-toxic, recyclable, and environmentally friendly compared to conventional solvents [68,91,96]. This coincides with a study by Jaffer et al. (2021) for the extraction of prodigiosin pigment that is commonly extracted by chloroform, which is renowned for its toxicity [97]. The application of an LBS with the use of polymer and salt as the phase-forming components has shown potential as an environmentally friendly alternative method to replace chloroform.

Furthermore, the conventional LLE strategy for recovery and extraction involves several steps, intricate routes, longer processing times, substantial energy inputs, and is also costly. The utilization of advanced LLE has proven to be simpler and faster; equilibrium distribution takes place at a rapid pace, low cost, and can be applied in a large-scale separation process. Sankaran et al. (2017) [75] reported that the fermentation using LBF, followed by the separation and purification of lipase from *B. cepacia*, resulted in a system that combines upstream and downstream processing in a single step, which accelerates product formation, improves product yield, makes downstream processing simpler and more cost-effective, and assists downstream processing.

Another difference between conventional and advanced LLE methods is the separation efficiency. Conventional LLE involves the development of an emulsion during the extraction of a specific aqueous sample due to the presence of surface-active chemicals on some natural materials. These surface-active compounds will adsorb at the liquid–liquid interface, resulting in the formation of an emulsion. The phase-forming components in the advanced LLE method comprise a considerable volume of water while maintaining a low interfacial layer between the two phases [91,98]. This is advantageous for separating proteins from cellular debris or purifying the targeted protein from contaminated proteins, whereas conventional LLE is mostly used to purify and separate biomolecules that are ion-sensitive, as most conventional LLE methods have a low-ionic system [91].

The LBS possessed very low interfacial tension between 0.0001 and 0.1 dyne/cm compared to the high interfacial tension between 1 and 20 dyne/cm for a conventional water–organic solvent system, which will consequently create a high interfacial contact area of the dispersed phases, thus providing efficient mass transfer [97,99]. The low interfacial tension and high water content of the advanced LLE method are also beneficial to preserving the activity of biomolecules in addition to improving the mass transfer of molecules [100].

### 2.5. Parameters in LBS

An LBS is adapted to the physicochemical parameters of the biomolecule’s solute for optimization purposes [96]. A study on the LBS stated that three factors are significant in enhancing protein recovery: pH, salt concentration, and temperature [101].

#### 2.5.1. pH System

The pH of an LBS can affect the extraction of specific compounds. The pH can change the charge and surface characteristics of the solute, hence disrupting biomolecule partitioning [96]. This occurs due to the alteration in the charge and solute characteristics. When the pH is greater than the isoelectric point, the net charge of the target biomolecules will become negative, and the net charge will become positive when the pH of the system is lower than the isoelectric point. Previous studies determined that the partitioning coefficient increases when the dipole moment is positive due to the higher pH of the system. This condition favors negatively charged target biomolecules toward the polymer-rich top phase [91].

#### 2.5.2. Molecular Weight of Polymer

The degree of hydrophobicity of the target biomolecule partitioning is affected by molecular weight. Hydrophobicity increases as the molecular weight of the polymer increases due to the presence of long hydrocarbon chains in monomers. The polymer-rich top phase’s free volume decreases due to the hydrophobicity of the system that divides the target biomolecules to the bottom phase. Hydrophobicity is low when the molecular weight is low; hence, the purification factor drops, and the target biomolecules are partitioned with the contaminated proteins that stay in the polymer-rich top phase [91].

#### 2.5.3. Temperature

According to a review, the effect of temperature on liquid biphasic systems is highly complex because the system’s behavior varies with temperature. Most of the publications found that as the temperature rises, the two-phase area of the polymer–salt binodal curve expands, implying that the critical salt concentration required to generate an LBS decreases. The polymer–solvent interaction will decrease at increasing temperatures, hence reducing the solubility of the polymer in water [102].

In addition, the changes in temperature will affect partitioning through viscosity and density. Thus, the temperature of the LBS is vital in obtaining a precise partitioning effect in the experiments. In a polymer–polymer LBS with a lower polymer concentration, phase separation is commonly achieved at lower temperatures. However, at lower temperatures, the polymer–salt system has a reverse effect [96].

#### 2.5.4. Polymer Concentration

Every LBS involves at least one type of polymer for the separation process. Therefore, it is crucial to acknowledge the behavior of polymers in different conditions. Two conditions may be applied for the concentration of a polymer in water. The polymer may be either in a diluted or concentrated condition [100]. The density of the bottom dextran-rich phase increased as the polymer concentration increased, and the difference in density is linearly related to the tie-line length [35]. Moreover, increased polymer concentrations are related to the phase’s high density, refractive index, and viscosity.

#### 2.5.5. Salt Concentration

The salt concentration in an LBS affects the solubility and interaction of the target biomolecules. When the salt concentration increases, the water’s surface tension rises, resulting in an increased hydrophobic interaction between the protein and the water. Studies determined that when the salt concentration is high, the saturation will also increase, hence reducing the solubility of the target biomolecules [91]. The effect of salt additives is one of the interesting issues regarding the theoretical and practical view on the partitioning of biomolecules. Several studies reported the positive influence of chloride salts at different concentrations on the partitioning of protein using the LBS [103].

## 3. Conclusions

In conclusion, conventional methods have successfully extracted proteins from various agricultural wastes; however, some drawbacks have been identified. The LBS has been implemented to overcome those issues. To further improve the effectiveness of separation, the advanced methods of LBS have been introduced. The methods are effective and superior in obtaining the targeted biomolecules compared to the conventional method. Furthermore, the drawback of conventional methods for extraction and purification in terms of economic and complexity of the operation has been improved. The parameters affecting the LBS are the key factors enhancing enzyme recovery. Thus, appropriate parameters must be determined to obtain effective extraction of targeted proteins. Studies on the advanced LBS as a green and highly efficient bioseparation technology appear to be encouraging as a novel integration process in obtaining amylase due to the simplicity and effectiveness of the system. Hence, the implementation of advanced LBS for the purification of amylase from agricultural waste is a promising method for utilizing waste.

## Figures and Tables

**Figure 1 foods-10-02748-f001:**
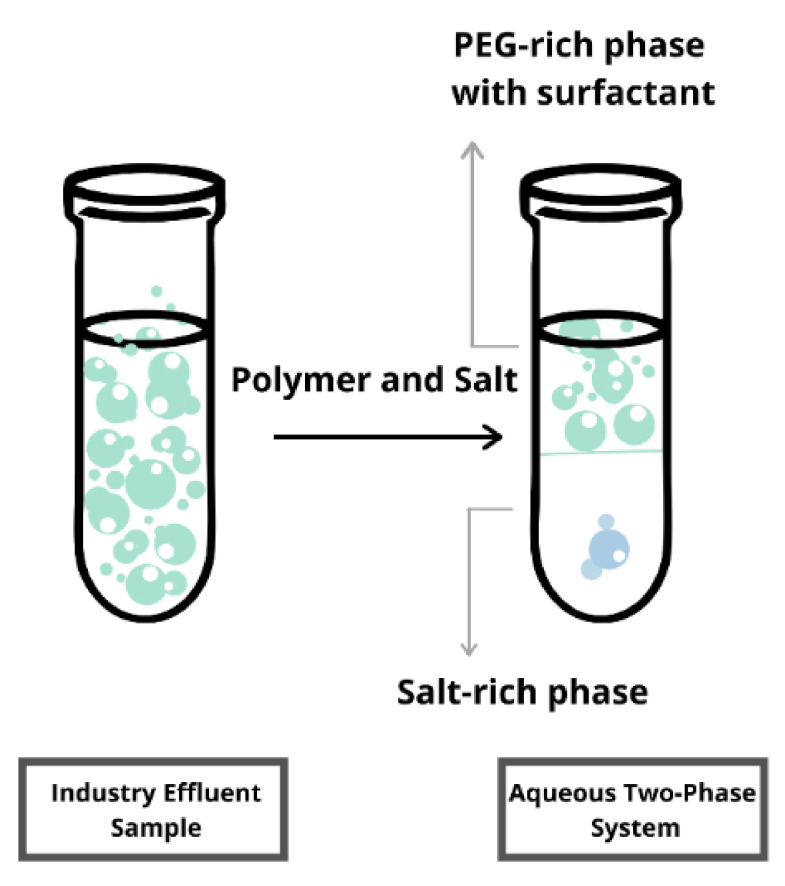
Graphical image of the liquid biphasic system.

**Figure 2 foods-10-02748-f002:**
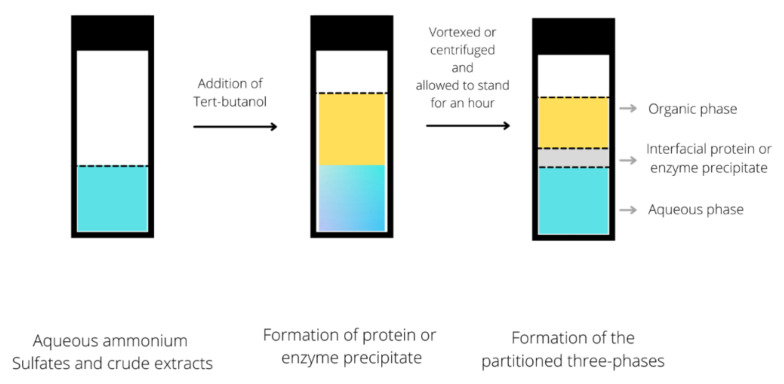
Schematic diagram of a three-phase partitioning system.

**Figure 3 foods-10-02748-f003:**
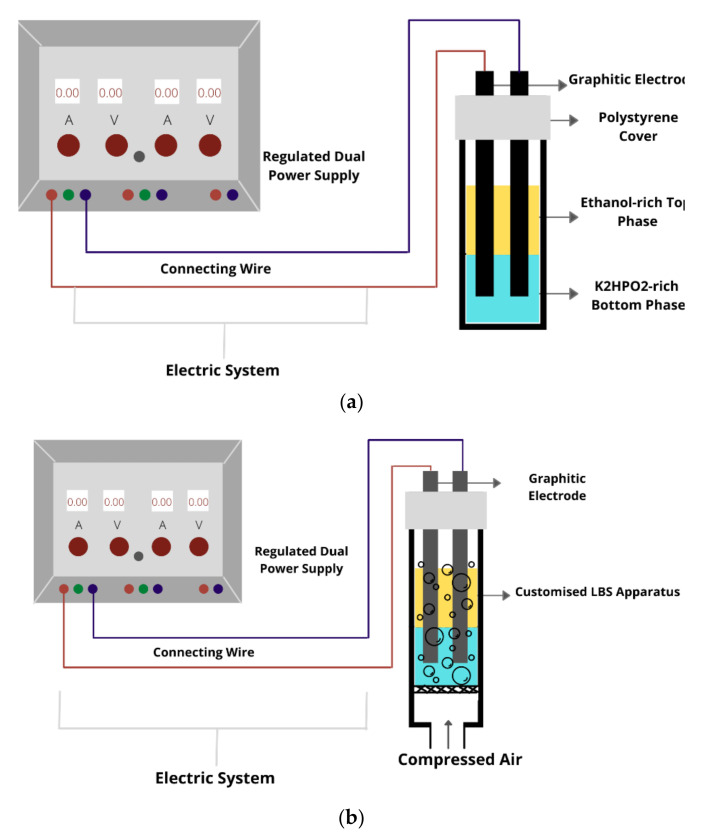
(**a**) The schematic diagram of the liquid biphasic electric partitioning system and (**b**) the liquid biphasic electric flotation system.

**Table 1 foods-10-02748-t001:** The classification of amylases and their applications.

Classification of Amylases	Alternative Names	Applications
Alpha-amylase	α-1,4-glucan-glucanohydrolase; EC 3.2.1.1	Degrades the α-1,4-glycosidic linkage of starch by breaking down starch to oligosaccharides or saccharides
Beta-amylase	1,4-D-glucan maltohydrolase; glycogenase; saccharogen amylase; EC 3.2.1.2	Catalyzes the hydrolysis of the second α-1,4-glycosidic linkage by cleaving the linkage from the non-reducing end
Gamma-amylase	Glucan-1,4-α-glucosidase; amyloglucosidase; exo-1,4-α-glucosidase; glucoamylase; lysosomal α-glucosidase; 1,4-α-D-glucan glucohydrolase	Breaks the α-1,6-glycosidic linkage and the last α-1,4-glycosidic linkage at the non-reducing end of amylopectin and amylose

**Table 2 foods-10-02748-t002:** The comparison between conventional and advanced LLE methods.

Criteria	Conventional LLE Methods	Advanced LLE Methods
Impact on the Environment	Requires the use of large volumes of highly pure solvents, where the reagents employed are no longer usable or recyclable, and thus, must be dumped into the environment, resulting in negative consequences to the environment	Phase-forming components are non-toxic and environmentally friendly compared to conventional solvents
Process Feasibility	Recovery and extraction involve several steps, intricate routes, longer processing time, substantial energy inputs, and are also costly	Simpler and faster, equilibrium distribution takes place in a short time, with low cost and the possibility to be applied in a large-scale separation process
Separation Efficiency	The development of an emulsion during the extraction of a specific aqueous sample is due to the presence of surface-active chemicals on some natural materials. These surface-active compounds will adsorb at the liquid–liquid interface, resulting in the formation of an emulsion	The phase-forming components in the advanced LLE method comprise a considerable volume of water while maintaining a low interfacial layer between the two phases
Interfacial Tension	High interfacial tension between 1 and 20 dyne/cm for a conventional water–organic solvent system	Low interfacial tension between 0.0001 and 0.1 dyne/cm

## Data Availability

Data is contained within the article.

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
