# Peer review of "Latest Advances in Protein-Recovery Technologies from Agricultural Waste"

_foods, 2021, doi:10.3390/foods10112748_

Round 1

Reviewer 1 Report

This work describes the extraction techniques of proteins from agricultural by-products. The topic of this review is interesting and revealing. Such type of work can have a significant contribution to the development of science in this field. Modes of information presentation are clear. However, the authors should let to edit the manuscript by a native English speaker (some lexical and stylistic errors). Furthermore, subsections: 2.3.5. Microwave-assisted LBS; 2.3.6. Ultrasound-assisted LBS; and 2.3.7. Electrical assisted LBS., authors should be extended by the description of studies concerning proteins extraction from agricultural waste - which is crucial for this manuscript. In the current version of the manuscript, the authors focused only on the extraction of plant secondary metabolites such as Resveratrol, piceid and emodin, vitexin, isovitexin, orientin (2.3.5.); Phenylethanoid glycosides (2.3.6.); betacyanin (2.3.7.) etc. Authors should add examples of the application of these mentioned techniques for proteins extraction (or replace current examples). It is strongly recommended to introduce information about the possible effects of ultrasounds, electric field, and microwaves on the protein structure and limitations of these techniques( 2.3.5; 2.3.6; 2.3.7) for proteins extraction (e.g. in section 2.4.). These issues should be improved.

Detailed revisions are given below (minor comments).

Line 69: replace “liberated” by “released”

Line 160: “Amid et al. (2014) [30] researched the extraction of amylase from dragon peel. The 161 dragon fruit is recognized as one of the world's leading commercial tropical fruits.” Move [30] at the end of phrase i.e.: “Amid et al. (2014) researched the extraction of amylase from dragon peel. The 161 dragon fruit is recognized as one of the world's leading commercial tropical fruits [30].” Please, correct it through the manuscript. (lines: 300, 307, 322, 587)

Line 425-426: add space after “Burkholderia” i.e. „Burkholderia cepacia”

Line 617: Plese, replace „NaCl” with „ionic strength” or „salt concentration”

Author Response

attached file

Reviewer 2 Report

The article reviews  the separation of proteins, with the main focus on amylase enzyme, from agricultural industrial waste using conventional and advanced techniques of   extraction and purification using a liquid biphasic system (LBS).

The topic of this paper in interesting and relevant, however, the article must be checked for English as it contains numerous mistakes. I have pointed out some but it is impossible to address all of them so It is recommended that the article is sent to English proof. Besides considerable changes must be mande to the structure so that the article is considerably improved.

The introduction should be restructures. Before authors state (line 49-50) that amylase will be the focus they should talk about other compounds/protein that are extracted from agriculture waste and then explain why they chose amylase/its importance. Lines 61-73 are a bit too specific to be in the introduction and should be placed elsewhere. Sentence from Line 75-79 is way to long and confusion. And many more?Be specific. From here one thinks that authors will talk about everything whereas from my understanding the focus in on  liquid biphasic system technology

2.1. Characteristics of Agricultural Waste and Enzyme: Authors do not talk about enzyme here. A statement should be added before it goes into further sub-divisions

In section 2.1.2

 It is recommended that authors provide a table with this information as it is easier to understand/visualize. Besides it makes more sense to first talk about the amyslase classification and then how/where it can be extracted from

The article repeats way too many times that Amylases are enzymes that break down starch by catalysing the hydrolysis

I strongly recommend that authors prepare a „quantitative “ table with some specific details of the different methods instead of just proving a „qualitative“ on (Table 1).

Some other minor comments:

Line 47: Localization of proteins?

Line 52: English check (for the broken down)

Line 55-56: If Previously then why is?

Line 59: English check

Line 97: Will be observed? By whom?

Line 104: Waste is a better term

Lines 104-106: Why so many „and“ its confusing

Line 243: English check

Table 1: Change necessitates for another word

Author Response

attached file

Round 2

Reviewer 1 Report

The current version of manuscript was deeply reviewed and English language corrections were made. However, the main issue of the previous review report was not addressed:

“Subsections: 2.3.5. Microwave-assisted LBS; 2.3.6. Ultrasound-assisted LBS; and 2.3.7. Electrical assisted LBS., authors should be extended by the description of studies concerning proteins extraction from agricultural waste - which is crucial for this manuscript. In the current version of the manuscript, the authors focused only on the extraction of plant secondary metabolites such as Resveratrol, piceid and emodin, vitexin, isovitexin, orientin (2.3.5.); Phenylethanoid glycosides (2.3.6.); betacyanin (2.3.7.) etc. Authors should add examples of the application of these mentioned techniques for proteins extraction (or replace current examples). It is strongly recommended to introduce information about the possible effects of ultrasounds, electric field, and microwaves on the protein structure and limitations of these techniques( 2.3.5; 2.3.6; 2.3.7) for proteins extraction (e.g. in section 2.4.). These issues should be improved to fulfil the basic requirements to be published in Foods.”

Please explain why these sections (2.3.5 – 2.3.7) were introduced into the manuscript if they are not concerning the extraction of the protein.

- underline the importance of the above-mentioned sections in the current version manuscript (write in manuscript).

- remove these sections if they are not important for the main issue of the manuscript - “Extraction and Purification of Protein from Agricultural Waste”

- correct it according to the previous outlines (the above mentioned previous review report

Author Response

We acknowledged the comment made by the reviewer that is very valuable in improving the quality of our manuscript.

On behalf of the authors, I would like to confirm that the manuscript has been revised again according to the comment. Kindly find enclosed our revised manuscript entitled “Latest Advances in Proteins Recovery Technologies from Agricultural Waste”. The correction has been marked up using the yellow highlight. Attached also is the response to the reviewer comment for your perusal.

I hope that you will be satisfied with the improvements made to the manuscript. I look forward to receiving your correspondence in due course.
